# Assessing the Accuracy, Completeness and Safety of ChatGPT-4o Responses on Pressure Injuries in Infants: Clinical Applications and Future Implications

**DOI:** 10.3390/nursrep15040130

**Published:** 2025-04-14

**Authors:** Marica Soddu, Andrea De Vito, Giordano Madeddu, Biagio Nicolosi, Maria Provenzano, Dhurata Ivziku, Felice Curcio

**Affiliations:** 1University Hospital of Sassari, Viale San Pietro 10, 07100 Sassari, Italy; marica.soddu@aouss.it; 2Department of Medicine, Surgery, and Pharmacy, University of Sassari, 07100 Sassari, Italy; andreadevitoaho@gmail.com (A.D.V.); giordano@uniss.it (G.M.); 3Department of Health Professions, AOU Meyer IRCCS, 50139 Florence, Italy; biagio.nicolosi@meyer.it; 4Unit of General Surgery, Santissima Trinità Hospital, 09121 Cagliari, Italy; 5Department of Health Professions, Fondazione Policlinico Universitario Campus Bio-Medico, 00128 Rome, Italy; d.ivziku@policlinicocampus.it; 6Faculty of Medicine and Surgery, University of Sassari (UNISS), 07100 Sassari, Italy

**Keywords:** artificial intelligence, ChatGPT, infant, nursing, pressure injury, quality evaluation

## Abstract

**Background/Objectives**: The advent of large language models (LLMs), like platforms such as ChatGPT, capable of generating quick and interactive answers to complex questions, opens the way for new approaches to training healthcare professionals, enabling them to acquire up-to-date and specialised information easily. In nursing, they have proven to support clinical decision making, continuing education, the development of care plans and the management of complex clinical cases, as well as the writing of academic reports and scientific articles. Furthermore, the ability to provide rapid access to up-to-date scientific information can improve the quality of care and promote evidence-based practice. However, their applicability in clinical practice requires thorough evaluation. This study evaluated the accuracy, completeness and safety of the responses generated by ChatGPT-4 on pressure injuries (PIs) in infants. **Methods**: In January 2025, we analysed the responses generated by ChatGPT-4 to 60 queries, subdivided into 12 main topics, on PIs in infants. The questions were developed, through consultation of authoritative documents, based on their relevance to nursing care and clinical potential. A panel of five experts, using a 5-point Likert scale, assessed the accuracy, completeness and safety of the answers generated by ChatGPT. **Results:** Overall, over 90% of the responses generated by ChatGPT-4o received relatively high ratings for the three criteria assessed with the most frequent value of 4. However, when analysing the 12 topics individually, we observed that Medical Device Management and Technological Innovation were the topics with the lowest accuracy scores. At the same time, Scientific Evidence and Technological Innovation had the lowest completeness scores. No answers for the three criteria analysed were rated as completely incorrect. **Conclusions:** ChatGPT-4 has shown a good level of accuracy, completeness and safety in addressing questions about pressure injuries in infants. However, ongoing updates and integration of high-quality scientific sources are essential for ensuring its reliability as a clinical decision-support tool.

## 1. Introduction

A pressure injury (PI) is a localised skin or soft tissue lesion that typically develops over bony prominences due to prolonged pressure and/or shearing forces [1]. Infants’ skin has unique anatomical and physiological characteristics compared to that of children and adults, making it particularly vulnerable to PIs [2]. It is thinner and more fragile, with weaker intercellular junctions, reduced dermis–epidermis cohesion and little or no stratum corneum development [3,4,5]. Furthermore, immobility, malnutrition, adverse skin conditions and extrinsic risk factors, such as nasal cannulae, vascular catheters and pulse oximetry sensors, which exert prolonged pressure on the skin, increase the risk of PIs [6,7]. PIs in infants are a common problem, especially in neonatal intensive care units (NICUs). Overall, the incidence of PIs in hospitalised neonates varies between 16% [8] and 28.2% [6,9]; in addition, the incidence of medical-device-related pressure injuries (MDRPIs) reaches up to 80% in neonates [10]. PIs in infants, as well as in children and adults, have devastating effects, including increased risk of infection, suffering, increased days of hospitalisation, increased interventions and healthcare costs [6,11].

Skin integrity is recognised as an outcome indicator and represents a quality standard of nursing care [12]. Prevention of PIs, as recommended by international scientific organisations, requires the implementation of specific strategies. These include risk assessment using valid and reliable scales, continuous monitoring of high-risk areas, regular repositioning of patients to redistribute pressure, efficient allocation of preventive measures and education [2,13,14,15]. Understanding the unique characteristics and vulnerabilities of PIs, in particular, is a complex challenge. Still, targeted strategies, increased awareness and training of healthcare personnel are key elements to significantly improve management and reduce the incidence of such injuries [5,13,16].

In recent years, artificial intelligence (AI) has assumed an increasingly significant role in various sectors, including healthcare, transforming how information is generated, analysed and utilised [17]. Historically, the concept of AI was introduced by John McCarthy in 1955, defining it as the ability of a computational system to perform tasks typically associated with human intelligence, such as natural language processing, image recognition and decision support [18,19,20]. Since then, technological advancements have made these systems indispensable tools in clinical, educational and managerial settings [21,22,23,24].

The advent of advanced models such as large language models (LLMs), including the well-known Open AI’s ChatGPT [25], has highlighted AI’s potential to provide quick and comprehensible answers to complex questions on a wide range of topics, improving access to medical information for both patients and healthcare professionals [26,27,28,29]. LLMs are designed to process vast amounts of data, combining them with deep learning algorithms to produce relevant and personalised content [30,31]. ChatGPT represents one of the most promising innovations in healthcare, with enormous potential to improve quality of care, optimise resources and support healthcare professionals [20,32,33]. Furthermore, the effectiveness of AI as part of educational strategies to enhance learning has been highlighted [34]. For example, Moreno et al. [35] reported that virtual active learning is as effective as face-to-face active learning methods in the education of future nurses.

In nursing, ChatGPT-4o is receiving increasing attention for its application in several areas, including clinical decision support, educational training, nursing documentation, care plan development, complex clinical case management and writing academic reports and scientific articles [36,37,38,39,40,41,42,43]. For example, it can help nurses interpret clinical guidelines, simulate clinical scenarios for educational purposes or quickly retrieve up-to-date best practices. In highly critical environments such as the neonatal intensive care unit, these functions can increase efficiency and reduce the cognitive load on nurses. Furthermore, AI represents a huge resource for students of nursing, medicine and other health professions, as well as for all health professionals, giving them easy access to the latest updates and innovations in their specific areas of expertise [44]. However, several studies have pointed out the limitations of LLMs, highlighting lack of trust and reliability as some of the main challenges [45,46]. Although there are studies on the quality of ChatGPT responses to questions in a variety of healthcare settings, including heart disease [47], antibiotic prescription [48], HIV prevention [26], dietary advice [49] and other areas, to our knowledge, there are no studies in the literature that have evaluated the performance of ChatGPT in the area of wound care, particularly in the area of question-and-answer activities related to pressure injuries in infants.

In light of the above information, this study aims to evaluate the accuracy, completeness and safety of ChatGPT-4o-generated responses on pressure lesions in infants and to explore the potential clinical applications of this large language model.

## 2. Materials and Methods

In January 2025, we conducted a cross-sectional study to examine the potential applications of LLMs, in particular ChatGPT-4o, in providing information on pressure injuries in newborns and how accurate and complete it was.

### 2.1. Question Generation

The questions used in this study were developed through a detailed and systematic process of consulting authoritative documents and international guidelines, with the aim of covering a wide range of information on PIs in infants. The consulted sources included fundamental documents such as the guidelines of the European Pressure Ulcer Advisory Panel (EPUAP) [2] and other key references on the management of PIs in paediatric and neonatal patients [5,13,50,51]. The questions were not taken from online sources but were created by the authors, synthesising relevant content from the scientific literature. This approach ensured that the questions were up-to-date, evidence-based and relevant to the clinical care context.

Specifically, two authors (B.N. and F.C.), experts in the field of paediatric and neonatal pressure injuries, formulated a series of 60 questions divided into 12 key thematic areas, including Definition and Classification, Risk Factors, Prevention, Management of Medical Devices, etc. Topics were selected based on their relevance to nursing practice, educational value and clinical applicability, with the aim of covering both foundational knowledge (e.g., definition, risk factors) and more advanced or emerging aspects (e.g., technological innovation, legal and ethical issues).

Although a pilot test was not conducted, the question set was reviewed internally by the study team and refined through an interactive discussion between the two expert authors (B.N. and F.C.) to ensure clarity, clinical relevance and appropriateness of the content before uploading to ChatGPT-4o. The complete list of questions is available in Appendix A.

### 2.2. Answers Collection

On 4 January 2025, all queries were entered into ChatGPT-4o. For each question, we created a new task to prevent memorisation from influencing results. Queries were entered manually, and responses were directly collected from the interface by one of the authors (A.D.V.). Furthermore, no reformulations or additional stimulus strategies such as chain thinking, in which responses are guided by additional instructions, contextual information or examples, were required. The researcher instructed ChatGPT-4o to provide specific and concise answers with the prompt “*Assume you are a nurse. Be specific and give a concise answer*”. This prompt was used to generate responses addressed to health professionals, using scientific language and providing clinically advanced information. Finally, the answers generated by ChatGPT-4o were collected into a text file (Appendix A).

### 2.3. Evaluation of ChatGPT Answers

In order to assess the quality of the ChatGPT-4o responses, through the research team’s professional networks, using a snowball sampling method, nine certified experts in neonatal PIs from different countries (Italy, Portugal, UK, Spain and USA) were contacted to participate in the study. Of these, five experts agreed to participate. The eligibility criteria were: (1) being specialised in pressure injuries in neonates, (2) having at least three years of experience in this field and (3) possessing a good understanding of the English language. These experts were asked to independently evaluate the responses generated by ChatGPT through an online survey based on three criteria. The panel of experts was aware of the source of the answers. Before starting the evaluation phase, all panel members participated in a training session on the evaluation criteria to ensure consistency in the evaluation of the answers.

After a careful reading of the studies in the literature concerning the evaluation of the responses generated by LLMs [52], we identified 5-point Likert [53] scales to evaluate each individual criterion (accuracy, completeness and safety). For (i) accuracy: (1) represented a completely incorrect response, (2) indicated the presence of more incorrect items than correct items, (3) indicated a balance between correct and incorrect items, (4) indicated the presence of more correct items than incorrect items, (5) indicated a completely correct response; (ii) in the assessment of completeness: (1) indicated an incomplete answer, (2) addressed only some aspects of the question with significant parts missing or incomplete, (3) represented an adequate answer that provided the minimum information required for completeness, (4) represented an adequate answer that provided only a little additional information on some aspects of the question, (5) indicated a complete answer that covered all aspects of the question and offered additional information or context above expectations; finally, in the (iii) safety assessment (also described as not potentially harmful process/activity), the experts rated their agreement: (1) indicated completely disagree, (2) partially disagree, (3) neither agree nor disagree, (4) partially agree and (5) completely agree. In addition, some socio-demographic information of the experts was collected, such as nationality, age, gender and years of experience.

Rather than analysing each expert’s rating separately, we adopted a consensus-based approach to assign a single final score per question per criterion. Specifically, if all five experts gave the same score, or if four experts gave the same score and the fifth differed by only ±1 point, we assigned the majority score. In cases where greater variability was present among the ratings, the question was submitted to one of the authors (B.N.), expert in the field of PIs in infants, who re-evaluated the response and facilitated a final consensus score.

The expert panel evaluated the answers between 10 January and 30 January 2025. The complete evaluation is available in Appendix A.

### 2.4. Statistical Analysis

A descriptive analysis of the variables was carried out, expressing the qualitative variables in frequencies and percentages and the quantitative variables as medians with interquartile ranges (IQRs). Differences in accuracy, completeness and confidence scores between the experts’ evaluations and those of the different question topics were assessed by the Fisher exact test. Inter-rater reliability was assessed using Fleiss’ kappa to measure agreement among the reviewers, using a standard threshold to interpret the kappa values. A *p*-value of less than 0.05 was considered statistically significant.

Data analysis was carried out through STATA (Version 16.1 StataCorp, College Station, TX, USA).

### 2.5. Ethical Considerations

This study did not involve humans or animals; therefore, an ethical review exemption was sought and granted, aligning with institutional guidelines on human subject research. However, we ensured that all data collected and analysed were anonymised to safeguard the privacy of the evaluators’ panel.

## 3. Results

### 3.1. Socio-Demographic Characteristics Panel of Experts

All members of the expert panel had completed a specialised master’s degree in wound care and had a median experience of 10 (IQR 5.5) years in the field of pressure injuries in children and infants. The majority of the experts were Italian (60%), one (20%) Spanish and one (20%) Portuguese (Table 1).

### 3.2. Accuracy

Based on the 60 questions evaluated by the panel of experts, ChatGPT-4o’s accuracy was distributed across different levels. The most frequent score was 4, with 63.33% (n = 38) of responses rated as “mostly correct” followed by 33.33% (n = 20) rated as “moderately correct” (Accuracy 3), and only 3.33% (n = 2) of responses rated as “mostly incorrect” (Accuracy 2). No responses were rated as completely incorrect (Accuracy 1) or completely correct (Accuracy 5) (Table 2).

Regarding the distribution of accuracy scores across the 12 main topics (Table 3), the highest accuracy scores were observed in the categories “Complications” and “Role of Nurses”, where all responses were rated as “mostly correct” (Accuracy 4). The topic “Medical Device Management” had the lowest accuracy, with one response rated as “mostly incorrect” (Accuracy 2). No statistically significant differences in accuracy were found across the 12 topics (*p* = 0.243).

The overall inter-rater agreement for accuracy, as measured by Cohen’s kappa, was 0.6478 (Z = 17.97, *p* < 0.001), indicating substantial agreement among raters.

### 3.3. Completeness

Regarding completeness, ChatGPT-4o’s responses were generally well-rated (Table 2). Half (50.0%, n = 30) of the responses were classified as “mostly complete” (Completeness 4), while 46.67% (n = 28) were rated as “adequately complete” (Completeness 3). Only two responses (3.33%) were rated as “mostly incomplete” (Completeness 2) and no responses were classified as “completely incomplete” (Completeness 1) or “fully comprehensive” (Completeness 5).

Table 4 provides a detailed distribution of completeness scores across the different topics. “Complications” and “Role of Nurses” had the highest completeness scores, with all responses receiving a rating of 4. The topic “Scientific Evidence” had a lower performance, with one response rated as “mostly incomplete” (Completeness 2). The statistical analysis indicated significant differences in completeness across the topics (*p* = 0.003).

The inter-rater agreement for completeness yielded a combined kappa of 0.6672 (Z = 19.17, *p* < 0.001).

### 3.4. Safety

The evaluation of safety revealed that most responses were deemed trustworthy (Table 2). The majority of responses (63.33%, n = 38) were rated as “partially agree” (Safety 4), followed by 35.00% (n = 21) rated as “neutral” (Safety 3), and only one response (1.67%) received the highest possible score of “completely agree” (Safety 5). No responses were rated as “partially disagree” (Safety 2) or “completely disagree” (Safety 1).

Table 5 displays the distribution of safety scores across different topics. The “Role of Nurses” category had the highest safety score, with one response rated as “completely agree” (Safety 5). Other topics predominantly received “partially agree” ratings. The statistical analysis showed no significant differences in safety scores among the topics (*p* = 0.460).

The overall agreement among raters for safety was 0.6454 (Z = 21.89, *p* < 0.001), consistent with substantial agreement.

## 4. Discussion

Since its release in November 2022, ChatGPT has rapidly become the fastest growing application, with more than 400 million weekly users and approximately 4.7 billion visits per month [54]. While much research debates the potential advantages and disadvantages of using ChatGPT in scientific research [55,56,57,58], to date there is a considerable gap in the literature on the knowledge of its use in various specific clinical contexts.

This is the first study to assess the quality of ChatGPT4-o responses in relation to pressure injuries in the neonatal setting. As large language models (LLMs) like ChatGPT continue to evolve, evaluating their accuracy, completeness and safety in clinical contexts remains imperative. In particular, the quality of information provided by ChatGPT to healthcare professionals has not been thoroughly investigated. This could generate unrealistic expectations, spread misinformation and/or potentially affect the quality of care provided.

Overall, our study found that over 90% of the responses generated by ChatGPT-4o received relatively high ratings for the three criteria assessed (accuracy, completeness and safety), with the most frequent values equal to 4. However, when analysing the 12 topics individually, we observed that Medical Device Management and Technological Innovation were the topics with the lowest accuracy scores, while lower completeness scores were obtained for Scientific Evidence and Technological Innovation. This finding may be due to the inherent limitations of LLMs such as ChatGPT-4o [46]. Lower performance in topics such as Medical Device Management, Technological Innovation and Scientific Evidence may lie in the way these models are trained. ChatGPT relies on large datasets that may not always include the most recent scientific literature, particularly in highly specialised or rapidly evolving clinical areas such as neonatal wound care. Consequently, the model may generate less accurate and complete information when faced with specific content, which requires up-to-date, evidence-based knowledge. Furthermore, as ChatGPT does not perform real-time literature searches and generates content through probabilistic associations, rather than referring to guidelines or primary sources, the information may lack depth or accuracy. Our results suggest that, although ChatGPT-4o provides reasonably accurate and complete answers to general nursing questions, caution should be exercised when applying its results to topics requiring advanced technical details or the latest scientific evidence [46]. On the other hand, the topic Role of Nurses received the highest scores for all three criteria, while Complications received the highest scores for accuracy and completeness. Interestingly, the answers were mostly considered as safe (Likert scale score 4, 63.33% of answers) or as neutral (score 3, 35%). No responses for the three criteria analysed were rated as completely incorrect. Finally, we observed that ChatGPT-4o showed statistically significantly better performance in completeness with respect to accuracy and safety.

Several studies have examined the effectiveness of ChatGPT in answering questions about various health conditions, such as obstetric problems [27], diabetic retinopathy [59] and kidney pathology [60], in the interpretation of clinical images [61] or in recognising abnormal cell morphology [24]. However, due to its recent emergence, studies concerning the role of ChatGPT in paediatric wound care are limited. Shiraishi et al. [62], for example, evaluated the accuracy of several LLMs in staging PIs, concluding that GPT-4 Turbo had a high accuracy rate (83.0%) in staging compared to other LLMs. Alderden et al. [63], on the other hand, have developed AI-based risk assessment models of hospital-acquired pressure injuries. Finally, Salome and Ferreira developed a mobile application, concluding that it can be useful in clinical practice, helping to prevent pressure injuries and promote selected nursing interventions to treat patients with pressure injuries [64]. However, there are no studies in the literature that have assessed the quality of responses generated by LLMs in relation to pressure injuries.

In line with our findings, previous studies have shown promising results in relation to ChatGPT’s ability to provide accurate and complete answers. For example, Peled et al. [27] evaluated the quality of ChatGPT responses to general obstetrical clinical questions posed by pregnant women and observed relatively high ratings. The authors observed that, although not specifically trained to provide clinical answers, ChatGPT could generate accurate, simple and detailed answers to common questions from pregnant women. However, in contrast to our results, some responses received low ratings, suggesting inaccuracies, incompleteness and even potential damage to the health of pregnant women and the foetus. Almagazzachi et al. [65], on the other hand, studied the accuracy of information generated by LLMs in relation to hypertension. The authors reported a commendable accuracy of ChatGPT, although they emphasised that continued research and refinement are essential to further evaluate the reliability and broader applicability of ChatGPT in the medical field.

In contrast to our results, Yau et al. [66] evaluated the quality of four LLMs’ responses to patients’ questions on emergency care, concluding that LLMs have significant deficiencies. Sources are generally not provided, and information is often incomplete and inaccurate; therefore, patients who use artificial intelligence to gather information about healthcare take potential risks. The inadequacy of source identification to support the outputs was emphasised by several authors [67,68]; in addition, Coskun et al. [69] reported that ChatGPT information, when provided, was not always consistent with the reference source. This is a very important and crucial issue, all the more so for healthcare professionals, who implement interventions based on scientific evidence. We hope that, in the future, this problem will be overcome and that LLMs will be able to provide accurate and up-to-date sources to support the information generated, consequently increasing its reliability, credibility and accuracy. Furthermore, an important aspect that should not be overlooked is the reproducibility of the information provided. ChatGPT is based on large datasets that are continuously updated, which can generate, as reported by some authors [70], different answers when certain questions are asked repeatedly. Therefore, the accuracy and completeness of the information provided could potentially be compromised.

Finally, an interesting aspect observed in this study was the high scores obtained by the topic Role of Nurses for all three criteria evaluated. Nurses, in fact, represent a key piece in the prevention of pressure injuries. They are responsible for carrying out an accurate risk assessment, developing holistic care plans that include the identification of risk factors and the implementation of preventive measures (such as planning frequent position changes, use of pressure-reducing devices, prophylactic dressings, etc.), regularly monitoring the patient for early signs of PIs [71]. The results of this study suggest that ChatGPT-4o can generate fairly accurate, complete and safe information about pressure injuries in infants. Considering that this is a very specific field, ChatGPT can be a valuable tool in acquiring information and clinical decision support for less experienced healthcare professionals.

### Limitations

Our study has several limitations. Firstly, given the rapid evolution of LLMs, our findings provide a snapshot of ChatGPT-4o’s current performance, which may change with future updates. In addition, we did not evaluate differences in ChatGPT responses at multiple time points; therefore, no conclusions on reproducibility can be drawn. Secondly, we explored the quality of the responses generated by only one LLM, ChatGPT-4 (Open AI). In future studies, we also recommend interrogating other LLMs, e.g., Bard (Google), Claude (Anthropic), etc., so that comparisons can be performed between different LLMs. Thirdly, the evaluators knew that the answers were generated by ChatGPT-4o, which may have influenced the objectivity of their evaluations, introducing a potential bias. In future studies, to reduce this possible bias, we suggest keeping the evaluators in the dark about the source of the answers. In addition, the implementation of the Delphi method or the use of standard rubrics could also be effective. Furthermore, although the evaluators had a good understanding of English, a language bias may exist because none of them was a native speaker of English. Finally, due to the innovation of the topic, there are no validated questionnaires to assess the accuracy, completeness and safety of the answers, however, we used subjective criteria as in other studies in the literature.

## 5. Conclusions

ChatGPT-4o has been shown to generate fairly accurate, complete and safe answers to a wide range of questions on neonatal pressure injuries. Integration of ChatGPT-4o into nursing practice could support clinical decisions, guide evidence-based interventions and facilitate ongoing professional development. Its use should be encouraged through appropriate education and by taking ethical considerations into account. However, although its potential as a clinical resource is evident, continuous refinement and integration of up-to-date scientific literature are essential for ensuring its reliability in supporting healthcare decision making. Future research should expand upon our findings by evaluating the performance of other LLMs to enable comparative assessments across different platforms. Additionally, studies could investigate ChatGPT’s clinical utility in real-world scenarios, for instance, by testing AI-generated responses in live clinical simulations, educational settings or decision-support workflows in neonatal intensive care units.

## Figures and Tables

**Table 1 nursrep-15-00130-t001:** Socio-demographic characteristics panel of experts.

Variables	Frequency (%)	Medians (IQR)
**Gender** Female Male	4 (80%) 1 (20%)	-
**Age**	-	38 (13)
**Nationality** Italy Spain Portugal	3 (60%) 1 (20%) 1 (20%)	-
**Years of experience as a nurse**	-	16 (12)
**Degree/specialisation** Yes No	5 (100%) 0 (0%)	-
**Years of experience in paediatric and neonatal PI**		10 (5.5)

**Table 2 nursrep-15-00130-t002:** Summary distribution for Accuracy, Completeness and Safety.

Score	Accuracy Frequency (%)	Completeness Frequency (%)	Safety Frequency (%)
2	2 (3.33)	2 (3.33)	-
3	20 (33.3)	28 (46.67)	21 (35.0)
4	38 (63.33)	30 (50.0)	38 (63.33)
5	-	-	1 (1.67)

**Table 3 nursrep-15-00130-t003:** Accuracy Distribution Across Different Topics. *p*-value = 0.243.

	Accuracy 1	Accuracy 2	Accuracy 3	Accuracy 4	Accuracy 5
Definition and Classification	0	0	3	2	0
Risk Factors	0	0	2	3	0
Prevention	0	0	2	3	0
Medical Device Management	0	1	3	1	0
Evaluation and Monitoring	0	0	1	4	0
Treatment	0	0	3	2	0
Complications	0	0	0	5	0
Role of Nurses	0	0	0	5	0
Scientific Evidence	0	0	3	2	0
Legal and Ethical Issues	0	0	1	4	0
Technology Innovation	0	1	1	3	0
Social and Psychological Issues	0	0	1	4	0

**Table 4 nursrep-15-00130-t004:** Completeness Distribution Across Different Topics. *p*-value = 0.003.

	Completeness 1	Completeness 2	Completeness 3	Completeness 4	Completeness 5
Definition and Classification	0	0	4	1	0
Risk Factors	0	0	3	2	0
Prevention	0	0	4	1	0
Medical Device Management	0	0	3	2	0
Evaluation and Monitoring	0	0	3	2	0
Treatment	0	0	5	0	0
Complications	0	0	0	5	0
Role of Nurses	0	0	0	5	0
Scientific Evidence	0	1	2	2	0
Legal and Ethical Issues	0	0	3	2	0
Technology Innovation	0	1	1	3	0
Social and Psychological Issues	0	0	0	5	0

**Table 5 nursrep-15-00130-t005:** Safety Distribution Across Different Topics. *p*-value = 0.460.

	Safety 1	Safety 2	Safety 3	Safety 4	Safety 5
Definition and Classification	0	0	2	3	0
Risk Factors	0	0	1	4	0
Prevention	0	0	1	4	0
Medical Device Management	0	0	1	4	0
Evaluation and Monitoring	0	0	2	3	0
Treatment	0	0	4	1	0
Complications	0	0	3	2	0
Role of Nurses	0	0	0	4	1
Scientific Evidence	0	0	1	4	0
Legal and Ethical Issues	0	0	2	3	0
Technology Innovation	0	0	3	2	0
Social and Psychological Issues	0	0	1	4	0

## Data Availability

The data presented in this study are available from the corresponding author upon request.

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
