# Peer review of "Assessing the Accuracy, Completeness and Safety of ChatGPT-4o Responses on Pressure Injuries in Infants: Clinical Applications and Future Implications"

_nursrep, 2025, doi:10.3390/nursrep15040130_

Round 1

Reviewer 1 Report

Comments and Suggestions for Authors

  • Write in more detail about how the questions were developed through the consultation of documents.

  • Make it clearer how the five specialists were chosen and selected.

  • Provide more details about the Likert scale.

Let me know if you need any further adjustments!"I suggest including this reference in the discussion: Salome GM, Ferreira LM. Developing a Mobile App for Prevention and Treatment of Pressure Injuries. ADVANCES IN SKIN & WOUND CARE, VOL. 31 NO. 2, 2018."

Author Response

Responses to Comments from Reviewer 1

Response:

Thank you very much, we greatly appreciate your support. We believe that your thoughtful comments have greatly clarified, improved and strengthened the manuscript.

Write in more detail about how the questions were developed through the consultation of documents.

Response

Thank you very much for your valuable comment and suggestion regarding the development of the questions. In response, we have expanded the description of the process by which the questions were developed, providing more detailed information in the materials and methods section of the revised manuscript.

Specifically, we clarified that the 60 questions used in the study were not derived from online sources but were formulated by two experts in the field of neonatal and pediatric pressure injuries. These questions were based on a thorough consultation of authoritative documents and international clinical practice guidelines, including the 2019 guidelines from the European Pressure Ulcer Advisory Panel (EPUAP), the National Pressure Injury Advisory Panel (NPIAP), and the Pan Pacific Pressure Injury Alliance (PPPIA), as well as other key literature on the topic.

The questions were organized into 12 thematic areas (e.g., definition and classification, risk factors, prevention, medical device management, etc.), and their formulation was guided by their relevance to nursing practice and potential clinical impact.

Make it clearer how the five specialists were chosen and selected.

Response

We thank the reviewer for this valuable comment. We agree that further clarification was needed regarding the selection of the five experts involved in the evaluation process. To address this shortcoming, we have revised the text in the Materials and Methods section (2.3. Evaluation of ChatGPT responses), providing more details on the selection process.

We greatly appreciate your suggestions, which have helped to improve the clarity and transparency of the methodology employed.

Provide more details about the Likert scale.

Response

Thank you for your suggestion. In this new version of the manuscript, we have added more details regarding the Likert scale (lines 163-165).

Let me know if you need any further adjustments!"I suggest including this reference in the discussion: Salome GM, Ferreira LM. Developing a Mobile App for Prevention and Treatment of Pressure Injuries. ADVANCES IN SKIN & WOUND CARE, VOL. 31 NO. 2, 2018."

Response

Thank you for your comment. In this new version we have included the reference as you suggested. Specifically, we have added:

“Finally, Salome and Ferreira developed a mobile application, concluding that it can be useful in clinical practice, helping to prevent pressure injuries and promote selected nursing interventions to treat patients with pressure injuries  [64].

  Reviewer 2 Report

Comments and Suggestions for Authors

This study on the reliability of CHATCPT on pressure injuries in infants seems quite instructive. However, there are many aspects that need to be corrected.

Expand the introduction a little more in terms of the areas of use of ChatGPT-4 nurses. Discuss this issue in the abstract section and in the conlusion section. Also, your research questions are missing and the limitations of your research section is not included in the article, please add these sections. Your method section is very insufficient, please add more details. The way the research was implemented and the theoretical framework on which the research is based are quite insufficient, I recommend that you think about these a little and do some research.

Author Response

Responses to Comments from Reviewer 2

This study on the reliability of CHATCPT on pressure injuries in infants seems quite instructive. However, there are many aspects that need to be corrected.

Response:

Thank you very much, we greatly appreciate your support. We believe that your thoughtful comments have greatly clarified, improved and strengthened the manuscript.

Expand the introduction a little more in terms of the areas of use of ChatGPT-4 nurses. Discuss this issue in the abstract section and in the conlusion section.

Response

Thank you for this valuable suggestion. As you suggested, we have expanded the Introduction by providing a more detailed and concise overview of the potential applications of ChatGPT-4o in nursing practice, including its use in clinical decision-making, education, documentation, communication, and care planning. In addition, we have briefly discussed these applications in the Abstract and Conclusion sections.

Also, your research questions are missing and the limitations of your research section is not included in the article, please add these sections.

Response

Thank you very much for your comment. Although the limitations of the study were already present at the end of the discussion, as you suggested, we added the specific section "4.1. Limitations.". In addition, among the supplementary materials we have added a new table (Table S1) listing the 60 questions generated. For your ease of reference below is the table found in the supplementary materials.

Your method section is very insufficient, please add more details.

Response

Thank you very much for your comment. In this new version of the manuscript, as you suggested, we have added more detail in the methods section. You will find them in red.

Thank you again for your support.

The way the research was implemented and the theoretical framework on which the research is based are quite insufficient, I recommend that you think about these a little and do some research.

Response

Thank you very much for your valuable comment. Based on your and other reviewers' comments, after careful reading of other studies, we have revised the manuscript. We believe that your thoughtful comments have significantly clarified, improved, and strengthened the manuscript.

We trust that we have met your expectations, thank you again.

Reviewer 3 Report

Comments and Suggestions for Authors

In this paper, the authors evaluated the performance of ChatGPT in generating responses in a specific area pressure injuries in infants from three perspectives, accuracy, completeness, and safety. They prepared 60 questions that were divided into 12 topics, and then asked 5 experts to evaluate each of the responses. In addition to descriptive analyses, they also conducted the following tests:

  1. Test whether there was inter-rater disagreement.
  2. Test if the scores among different topics were different.

I think overall this paper was written clearly. The methods the authors used and the experiments that they conducted made scientific sense. A particular strength in my mind is the high data quality. They prepared well-defined clinical questions and have solid expert panel review.

Below are my comments and suggestions.

  1. I understand that the authors wanted to first test if scores from different experts were different and then used the scores that had been agreed upon to test if scores among different topics were different. But have they considered conducting the 2nd test directly? In other words, they could built a model with two factors, one is the expert and the other is topic. By doing so, they will have more samples/data points (5 * 60), and they have already accounted for inter-rater variability.
  2. As the authors mentioned, they did not evaluate differences in ChatGPT responses at multiple time points, and thus temporal variability was not considered.
  3. The authors could go beyond just comparing performance across topics or raters, and focused more explicitly on inference about ChatGPT’s overall performance, as I think that is what the authors and/or the audience care most. Here are a few ones I can think of. They could build a confidence interval for the mean or the median score. They could test whether the performance exceeds a clinically acceptable threshold (e.g., mean ≥ 4). And ideally they could compare the performance across models or against expert(human)-generated answers.

Author Response

Responses to Comments from Reviewer 3

In this paper, the authors evaluated the performance of ChatGPT in generating responses in a specific area pressure injuries in infants from three perspectives, accuracy, completeness, and safety. They prepared 60 questions that were divided into 12 topics, and then asked 5 experts to evaluate each of the responses. In addition to descriptive analyses, they also conducted the following tests:

Test whether there was inter-rater disagreement.

  1. Test if the scores among different topics were different.
  2. I think overall this paper was written clearly. The methods the authors used and the experiments that they conducted made scientific sense. A particular strength in my mind is the high data quality. They prepared well-defined clinical questions and have solid expert panel review.

Below are my comments and suggestions.

Response

Thank you very much, we greatly appreciate your support. We believe that your thoughtful comments have greatly clarified, improved and strengthened the manuscript.

  1. I understand that the authors wanted to first test if scores from different experts were different and then used the scores that had been agreed upon to test if scores among different topics were different. But have they considered conducting the 2nd test directly? In other words, they could built a model with two factors, one is the expert and the other is topic. By doing so, they will have more samples/data points (5 * 60), and they have already accounted for inter-rater variability.

Response

Thank you for your thoughtful suggestion. We appreciate the idea of modelling the data using both "expert" and "topic" as factors to leverage all available ratings and account for inter-rater variability.

However, in our study design, we aimed to assign a single, consolidated score per question to reflect the panel’s final judgment rather than treating each expert rating independently. Specifically, when all five experts agreed on a score, or when four experts gave the same score and the fifth differed by ±1 point, we used the majority score. In cases of greater disagreement, the item was re-reviewed and discussed by one of the authors (B.N.), an expert in the field, who helped reach a consensus score. This process ensured that the final score for each question represented a considered and clinically grounded evaluation, rather than relying on a purely statistical aggregation.

Given our goal to assess the overall perceived quality of each response, we felt this approach better captured the expert consensus and was more appropriate for comparing performance across clinical topics. Nevertheless, we recognize that modelling the full dataset with all individual ratings could offer additional insights, and we appreciate your suggestion as a valuable perspective for future work. We explained this better in the methods section.

  1. As the authors mentioned, they did not evaluate differences in ChatGPT responses at multiple time points, and thus temporal variability was not considered.

Response

We fully agree with your comment, assessing temporal variability is a very interesting aspect. We are working in this direction for a future study. Thank you

  1. The authors could go beyond just comparing performance across topics or raters, and focused more explicitly on inference about ChatGPT’s overall performance, as I think that is what the authors and/or the audience care most. Here are a few ones I can think of. They could build a confidence interval for the mean or the median score. They could test whether the performance exceeds a clinically acceptable threshold (e.g., mean ≥ 4). And ideally they could compare the performance across models or against expert(human)-generated answers.

Response

We appreciate this reflective and constructive feedback suggestion. We share the view that comparing responses generated by ChatGPT with those generated by experts would be informative and is an important line of investigation for the future. Such a comparative evaluation was outside the research purview of this study, though, as it was specifically focused on testing ChatGPT-4o's independent performance within a clinically meaningful scenario.

As for conducting proper inferential analyses like estimating the confidence intervals for the means or medians of scores or testing for deviation at a pre-specified clinical cutoff (e.g., ≥4), we value the degree of statistical rigor such methods would bring. However, our main objective was to undertake a descriptive but systematic evaluation of ChatGPT output based on aggregated expert judgment at the topic level, without formal hypothesis testing.

With this being, as far as we are aware, the initial study directly evaluating the accuracy, completeness, and safety of ChatGPT-4o for neonatal pressure injuries, our choice was for descriptive, topic-stratified analysis as being most appropriate to our aims — especially in an area in which no such prior baseline performance or clinical standards for LLMs are as yet established. However, we do feel that adding inferential metrics would be useful in future studies, such as those evaluating multiple models or changes in performance over time.

In cases where greater variability was present among the ratings, the question was submitted to one of the authors (B.N.), expert in the field of PIs in infants, who re-evaluated the response and facilitated a final consensus score.

Reviewer 4 Report

Comments and Suggestions for Authors

Dear authors, the manuscript presents a timely and original contribution by evaluating ChatGPT-4o's clinical utility in addressing pressure injuries in infants. The methodology is clearly described, and the use of expert evaluation enhances the rigor. However, some sections would benefit from a clearer structure, greater methodological detail, and a stronger critical discussion.

Here are my comments:

Introduction

  • Consider reducing repetition (e.g., LLM capabilities are discussed in multiple places).

  • You mention 4.7 billion ChatGPT users (line 246). Please verify or clarify the source.

Methods

  • Clarify how inter-rater disagreements were resolved. Did the two additional raters discuss and reach consensus, or were their ratings averaged?

  • Elaborate on whether the experts were blinded to the source of the answers. It’s mentioned in the limitations, but including it more explicitly in the methodology would strengthen the manuscript.

  • Provide more details about the question development process: How were the topics identified? Were any pilot tests conducted?

  • Explain why ChatGPT was prompted as a "nurse". Does this affect the structure or tone of the responses?

Results

  • Add standard deviation or interquartile range (IQR) to describe Likert scores more granularly.

  • The tables could benefit from visual aids to highlight topic performance at a glance.

  • In Table 2, consider aligning variables to improve readability.

Discussion

  • Discuss why ChatGPT may underperform in certain areas (e.g., technical details or use of up-to-date literature).

  • Be more critical regarding the lack of source citation in ChatGPT responses.

  • Include a brief reflection on reproducibility. Would running the same queries now produce the same results?

  • Expand on the limitation concerning the subjectivity of expert evaluation. Could this be minimized in the future using standard rubrics or Delphi methods?

  • Consider adding that language bias may exist if ChatGPT was queried in English but the experts are mostly non-native speakers.

Conclusion

  • Suggest specific next steps for research (e.g., evaluating other LLMs, testing in live clinical scenarios).

Technical and stylistic notes

  • Minor grammar corrections are needed throughout the paper.

  • Use consistent terminology throughout (e.g., ChatGPT-4o, LLMs, AI chatbot).

  • Ensure references are formatted consistently (some contain typos or spacing issues).

Author Response

Responses to Comments from Reviewer 4

Dear authors, the manuscript presents a timely and original contribution by evaluating ChatGPT-4o's clinical utility in addressing pressure injuries in infants. The methodology is clearly described, and the use of expert evaluation enhances the rigor. However, some sections would benefit from a clearer structure, greater methodological detail, and a stronger critical discussion.

Here are my comments:

Response

Thank you very much, we greatly appreciate your support. We believe that your thoughtful comments have greatly clarified, improved and strengthened the manuscript.

 Introduction

Consider reducing repetition (e.g., LLM capabilities are discussed in multiple places).

Response

Thank you for your valuable comment. In this new version, we have revised the introduction and reduced the repetitions as you suggested

You mention 4.7 billion ChatGPT users (line 246). Please verify or clarify the source.

Response

Thank you very much for your comment. We checked and indeed we had made an error in reporting the information. In the new version we have corrected it with "400 million weekly users”.

Methods

Clarify how inter-rater disagreements were resolved. Did the two additional raters discuss and reach consensus, or were their ratings averaged?

Response

Thank you very much for your comment. We apologise if we were unclear.

As reported in the statistical analyses, the agreement between the reviewers was assessed using Fleiss' Kappa. To avoid confusion among readers, in this new preferred to delete the sentence “In cases of disagreement among the initial raters, the final decision was made by two additional experts (F.C. and M.S) to ensure consensus”.

Elaborate on whether the experts were blinded to the source of the answers. It’s mentioned in the limitations, but including it more explicitly in the methodology would strengthen the manuscript.

Response

Thank you for your comment. We have added this information in the manuscript, as you suggested (lines 157-158).

Provide more details about the question development process: How were the topics identified? Were any pilot tests conducted?

Response

Thank you very much for your valuable comments and suggestions, which have contributed to improving the quality and clarity of our manuscript.

In response to your request for more details about the question development process, we have revised the “Materials and Methods” section (paragraph 2.1) to clarify how the topics were identified and to specify whether any pilot tests were conducted. In particular, we have added information regarding the selection of the twelve key topics—based on a review of authoritative clinical guidelines and literature—and we have clarified that, although no formal pilot test was conducted, the questions were reviewed and refined internally by two expert authors.

Explain why ChatGPT was prompted as a "nurse". Does this affect the structure or tone of the responses?

Response

Thank you for your comment. In this new version, as you suggested, we have added the reason behind the prompt used. Specifically, we have added:

“This prompt was used to generate responses addressed to health professionals, using scientific language and providing clinically advanced information”.

Results

Add standard deviation or interquartile range (IQR) to describe Likert scores more granularly.

Response

Thank you for the suggestion. We appreciate the importance of presenting data in a detailed and interpretable manner. However, as noted in the Methods section, the Likert-scale ratings used in our study were treated as categorical ordinal data, and not as continuous variables. For this reason, we reported the absolute and relative frequencies of each score, which is the most appropriate descriptive approach for Likert data when not assuming equal intervals between categories.

While measures such as means or standard deviations are sometimes used in large-scale survey research where Likert scales are treated as approximations of continuous variables, we chose not to adopt this approach in order to remain consistent with established guidelines for the analysis of ordinal data.

The tables could benefit from visual aids to highlight topic performance at a glance.

Response

We thank the reviewer for the suggestion. The current formatting of the tables follows the journal’s author guidelines for presenting categorical data. However, should the editorial team consider the inclusion of visual aids (e.g., color shading or graphical enhancements) appropriate, we would be happy to revise the tables accordingly to improve readability and highlight topic performance more intuitively.

In Table 2, consider aligning variables to improve readability.

Response

Thank you for your valuable comment, we have revised the table according to your suggestion. We believe that its clarity and readability has improved considerably

Discussion

Discuss why ChatGPT may underperform in certain areas (e.g., technical details or use of up-to-date literature).

Response

Thank you very much for this insightful observation, which allowed us to improve the quality and depth of the Discussion section. We agree that understanding the limitations of ChatGPT is crucial, especially when applied to highly specialised clinical topics.

In the discussion section, we briefly explain that ChatGPT-4o may underperform in topics such as Technological Innovation and Medical Device Management due to the structural limitations of large language models. Specifically, we have added (lines 294-307):

“This finding may be due to the inherent limitations of LLMs such as ChatGPT-4o [46]. Lower performance in topics such as Medical Device Management, Technological In-novation and Scientific Evidence may lie in the way these models are trained. ChatGPT relies on large datasets that may not always include the most recent scientific literature, particularly in highly specialised or rapidly evolving clinical areas such as neonatal wound care. Consequently, the model may generate less accurate and complete information when faced with specific content, which requires up-to-date, evidence-based knowledge. Furthermore, as ChatGPT does not perform real-time literature searches and generates content through probabilistic associations, rather than referring to guidelines or primary sources, the information may lack depth or accuracy. Our results suggest that although ChatGPT-4o provides reasonably accurate and complete answers to general nursing questions, caution should be exercised when applying its results to topics requiring advanced technical details or the latest scientific evidence [46].”

Be more critical regarding the lack of source citation in ChatGPT responses.

Response

Thank you very much for your comment. As suggested, we have expanded this in the discussion section. Specifically, we have added (lines 344-350):

“The inadequacy of source identification to support the outputs was emphasised by several authors [67,68]; in addition, Coskun et al. [69] reported that ChatGPT information, when provided, was not always consistent with the reference source. This is a very important and crucial issue, all the more so for healthcare professionals, who implement interventions based on scientific evidence. We hope that, in the future, this problem will be overcome and that LLMs will be able to provide accurate and up-to-date sources to support the information generated, consequently increasing its reliability, credibility and accuracy.”

Include a brief reflection on reproducibility. Would running the same queries now produce the same results?

Response

Thank you very much for your comment. As suggested, we have included a brief reflection in the Discussion section. Specifically, we have added:

“Furthermore, an important aspect that should not be overlooked is the reproducibility of the information provided. ChatGPT is based on large datasets that are continuously updated, which can generate, as reported by some authors [70], different answers when certain questions are asked repeatedly. Therefore, the accuracy and complete-ness of the information provided could potentially be compromised.”

Expand on the limitation concerning the subjectivity of expert evaluation. Could this be minimized in the future using standard rubrics or Delphi methods?

Response

Thank you very much for your valuable comment. We have expanded on this limitation, as you suggested, in section 4.1 limits. Specifically, we have added:

“In future studies, to reduce this possible bias, we suggest keeping the evaluators in the dark about the source of the answers. In addition, the implementation of the Delphi method or the use of standard rubrics could also be effective.”

Consider adding that language bias may exist if ChatGPT was queried in English but the experts are mostly non-native speakers.

Response

Thank you very much for your suggestion. The authors fully agree. In this new version, in the limits section, we have added:

“Furthermore, although the evaluators had a good understanding of English, a language bias may exist because none of them were native speakers of English.”

Conclusion

Suggest specific next steps for research (e.g., evaluating other LLMs, testing in live clinical scenarios).

Response

Thank you very much for this insightful observation. As you suggested, we have added it in the conclusion section. Specifically, we have added:

“Future research should expand upon our findings by evaluating the performance of other LLMs to enable comparative assessments across different platforms. Additionally, studies could investigate ChatGPT’s clinical utility in real-world scenarios, for instance by testing AI-generated responses in live clinical simulations, educational settings, or decision-support workflows in neonatal intensive care units.”

Technical and stylistic notes

Minor grammar corrections are needed throughout the paper.

Response

Thank you very much for your comment. We have revised the manuscript and corrected the grammatical errors present

Use consistent terminology throughout (e.g., ChatGPT-4o, LLMs, AI chatbot).

Response

Thank you for your suggestion. We have revised the text and standardised terminology

Ensure references are formatted consistently (some contain typos or spacing issues).

Response

Thank you for your comment. We have reviewed and corrected the formatting of the references.

Round 2

Reviewer 2 Report

Comments and Suggestions for Authors

Publish, thanks for revised...